# Evaluation of Midwives’ Practises on Herpetic Infections during Pregnancy: A French Vignette-Based Study

**DOI:** 10.3390/healthcare11030364

**Published:** 2023-01-28

**Authors:** Adrian Mrozik, Yann Sellier, Déborah Lemaitre, Laurent Gaucher

**Affiliations:** 1Obstetric Department, Hospital Group Paris Saint Joseph, 75014 Paris, France; 2French College of Midwives (Collège National des Sages-Femmes de France, CNSF), 75010 Paris, France; 3EA 7328, Fetal Medicine Department Necker Hospital France, AP-HP, 92150 Suresnes, France; 4School of Maieutics of Foch, UVSQ, 78180 Montigny-le-Bretonneux, France; 5Public Health Unit, Hospices Civils de Lyon, 69500 Bron, France; 6INSERM U1290, Research on Healthcare Performance (RESHAPE), Claude Bernard Lyon 1 University, 69008 Lyon, France; 7Geneva School of Health Sciences, HES-SO University of Applied Sciences and Arts Western Switzerland, 1206 Geneva, Switzerland

**Keywords:** genital herpes, pregnancy, midwife, primary initial infection, nonprimary initial infection, recurrence, guidelines, neonatal herpes

## Abstract

(1) Background: One out of two pregnant women has a history of herpes infection. Initial infections have a high risk of neonatal transmission. Our objective was to analyse the professional practises of midwives regarding the management of herpes infections during pregnancy in France; (2) Methods: A national survey conducted via an online self-questionnaire, including clinical vignettes for which the midwives proposed a diagnosis, a drug treatment, a mode of birth, and a prognosis. These responses were used to evaluate the conformity of the responses to the guidelines, as well as the influence of certain criteria, such as mode of practise and experience; (3) Results: Of 728 responses, only 26.1% of the midwives reported being aware of the 2017 clinical practise guidelines. The midwives proposed taking the appropriate actions in 56.1% of the responses in the case of a recurrence, and in 95.1% of the responses in the case of a primary infection. For the specific, high-risk case of a nonprimary initial infection at 38 weeks of gestation, reporting knowledge of the recommendations improved the compliance of the proposed care by 40% (*p* = 0.02). However, 33.8% of the midwives underestimated the neonatal risk at term after a primary initial infection, and 43% underestimated the risk after a primary initial infection at term; (4) Conclusions: The majority of reported practises were compliant despite a low level of knowledge of the guidelines. The dissemination of guidelines may be important to improve information and adherence to appropriate therapeutic practise.

## 1. Introduction

According to the World Health Organization (WHO), genital herpes affects more than 500 million people in the world [1]. Herpes simplex virus (HSV) infection is often nonsevere, except in special cases, such as newborns or immunocompromised individuals [2,3]. One out of two pregnant women has a history of herpes infection, most often with HSV1 [4]. Neonatal herpes occurs in 10 per 100,000 births, which represents approximately 14,000 cases per year internationally. The severity of neonatal herpes is due to the multitude of complications. Indeed, without treatment, mortality reaches 60 to 70% of cases [5].

In 2017, the National College of Obstetrician Gynaecologists of France (CNGOF) published the first guidelines in France for good practise concerning the medical care of pregnant women infected with a herpes simplex virus [6]. These guidelines distinguish three contexts of herpes simplex infections: primary initial infection, nonprimary initial infection, and recurrent infection. A primary initial infection is the first contact with HSV. A nonprimary initial infection is the first contact with one of the two specific serotypes of HSV after the other serotype has already been encountered. Finally, a recurrence is defined as a reactivation of a previously encountered serotype. The type of maternal infection, whether primary initial, nonprimary, or recurrent, does not influence the severity of neonatal herpes if the child is infected. However, the risk of transmission is different for each type of maternal infection. An initial infection, despite whether it is primary or nonprimary, has a higher risk of transmission than does a recurrence [4]. For an initial infection during labour, the risk of neonatal herpes varies between 25 and 57%, depending upon whether it is a nonprimary initial or a primary initial infection, respectively [7]. In comparison, for a recurrence, maternal–foetal transmission in the peripartum period is observed in only 1% of cases [7]. One explanation for these high transmission rates is the inability and/or the difficulty of preventing these infections. Indeed, 70% of neonatal herpes diagnoses are of children born to asymptomatic mothers with no history of herpes [8]. It is common to diagnose an initial infection after the critical period of birth, in the postpartum period, despite the necessity of treating it earlier [9].

A lack of knowledge about the prevention and management of herpes during pregnancy was highlighted by a study conducted just before the publication of the latest French guidelines [10]. For instance, 59% of the gynaecologists–obstetricians and 43.5% of the midwives questioned responded that they do not systematically prescribe prophylaxis at the end of pregnancy in the context of an initial infection. This is even more alarming as the interest in systematically proposing this prophylaxis has been demonstrated by a meta-analysis published by the *Cochrane Database* in 2008 [11]. In this publication, it is reported that women who received antiviral prophylaxis were significantly less likely to have a recurrence of genital herpes at delivery (relative risk (RR) 0.28, 95% confidence interval (CI) 0.18 to 0.43). Women who received antiviral prophylaxis were also significantly less likely to have a caesarean delivery for genital herpes (RR 0.30, 95% CI 0.20 to 0.45). Women who received antiviral prophylaxis were significantly less likely to have HSV detected at delivery (RR 0.14, 95% CI 0.05 to 0.39) [11].

In France, midwives are healthcare professionals in primary care; they can diagnose diseases; they can prescribe every medical test without any restrictions; and they can prescribe drugs to prevent herpes recurrence. Considering that, in 2017, less than half of midwives were prescribing the recommended prophylactic treatment, it seemed essential to evaluate their practises 3 years later in order to consider possible ways of improvement.

Therefore, we proposed to evaluate midwives’ practises in the medical care of pregnant women with herpes simplex virus infections. Our main objective was to estimate the proportion of French midwives who were aware of the recommendations. Our secondary objectives were to estimate the proportion of midwives who made a correct diagnosis, who proposed the correct treatment, and who correctly assessed the risks for the newborn.

## 2. Materials and Methods

We conducted a nationwide, voluntary, open e-survey to recruit a convenience sample of independent midwives working in France during the first COVID-19 lockdown. This quantitative study followed the *Checklist for Reporting Results of Internet E-Surveys* (CHERRIES) in the reporting of our data (see Appendix A) [12].

The link to the survey was disseminated by e-mail from November 2020 to March 2021 to all French midwives.

According to the *Direction de la recherche, des études, de l’évaluation et des statistiques*, 13,801 hospital midwives were practising in France on 1st January 2021 (https://drees.shinyapps.io/demographie-ps/; accessed on 20 January 2023). This questionnaire was sent via the e-mail accounts of the heads of the maternity units. In addition, it was distributed through the newsletter of December 2020 of the French College of Midwives (Collège National des Sages-Femmes de France, CNSF). A nonprobability chain-referral sampling was obtained via e-mail. All midwives gave their consent before participating in the study, as described in the ethics approval section. Participation was voluntary, without any incentive or reward.

Participants provided informed consent by participating in the study. Participation was anonymous and participants could stop at any time by leaving the website. The beginning of the questionnaire clearly stated the objectives of the study, the estimated length of time that completing the survey would take, where and for how long data would be stored, who the investigators were, the Ethics Committee registration number, and the procedure for submitting objections to such research to the national authorities. This study was approved by the Ethics Committee of Foch Hospital (decision n°IRB00012437).

The questionnaire was constructed by the authors to contain three parts. The first part of the survey questioned the participants about their socio-demographic characteristics (age, gender, mode of practise, department of practise, duration of practise, and number of prenatal visits conducted per week). The second part assessed the midwives’ theoretical knowledge of the CNGOF guidelines. The third part, composed of clinical vignettes, was used to evaluate the midwives’ practise (see the Appendix A). We created 6 vignettes, with vignette #1 corresponding to an primary initial infection at 25 weeks of gestation, vignette #2 to an nonprimary initial infection at 38 weeks, vignette #3 to a recurrence at 38 weeks, vignette #4 to an nonprimary initial infection at 25 weeks, vignette #5 to a recurrence at 25 weeks, and vignette #6 to an primary initial infection at 38 weeks (see Appendix A).

The questionnaire was tested on five midwives to verify the items’ usability, technical functionality, clarity, and reliability. It was then administered with SurveyMonkey^®^ software. Entering a response to all of the items was mandatory. Questions were neither randomised nor alternated. There were no more than 10 items per page so as to avoid discouraging the respondent and to improve the completion rate of the survey. The questionnaire is available in Appendix A. We did not use cookies. The IP addresses registered by SurveyMonkey^®^ were not extracted.

Only completed questionnaires were analysed. All statistical analyses were performed with R software, version 4.0.3. Quantitative variables were expressed as means and standard deviations (SD), and then compared using Welch’s two-sample t-test, or as medians [25–75th percentiles] and then compared using a Wilcoxon Rank-Sum Test, according to their distributions. Qualitative variables were expressed as counts (percentages) and then compared using Fisher’s exact test.

## 3. Results

Complete responses were received from 728 midwives (Figure 1). Among them, 371 (51.0%) reported working in medium-sized maternity units; the median reported age was 34 years old; 361 (50.0%) reported practising in one of the two largest French metropolitan areas, those being Paris and Lyon (Table 1).

Among these midwives, 26.1% declared that they were familiar with the guidelines (Table 1). In terms of midwifery practises/skills, diagnosis had the highest rate of correct answers, except for the nonprimary initial infection scenarios. The proportion of correct diagnoses were, respectively, 95.2% at 25 weeks of gestation, and 92.2% at 38 weeks of gestation for the scenarios including primary initial infections (Table 2). For the nonprimary initial infection scenarios, the proportions of correct diagnoses were, respectively, 57.0% at 25 weeks of gestation, and 47.4% at 38 weeks of gestation (Table 3). The proportions of correct diagnoses for recurrent infections were, respectively, 88.8% at 25 weeks of gestation, and 90.5% at 38 weeks of gestation (Table 4). 

For the clinical vignettes including women at term, the proposed drug treatment was appropriate in more than half of the cases (i.e., 76.2% for primary initial infections, 68.8% for nonprimary initial infections, and 56.1% in recurrences; Table 2, Table 3 and Table 4). However, the mode of delivery proposed by the midwives was mostly inappropriate (i.e., 50.0% for primary initial infections, 64.7% for initial nonprimary infections, and 75.8% for recurrences; Table 2, Table 3 and Table 4).

Knowledge of guidelines was found to be associated with a statistically significant improvement in care (i.e., appropriate drug treatment associated with appropriate mode of birth), especially in the specific, high-risk case of an nonprimary initial infection at 38 weeks of gestation, where reporting knowledge of the recommendations improved the compliance of the proposed care by 40%, from 21.7 to 30.5% (*p*-value = 0.019; Appendix A). Moreover, we observed that knowledge of the guidelines improved the diagnosis in the specific case of nonprimary initial infections, i.e., from 44.1 to 56.8%, *p* = 0.003 (Appendix A).

In the case of a primary initial infection at 38 weeks of gestation, 48.6% of the midwives underestimated the neonatal risk. For a nonprimary initial infection, the proportion of the midwives who underestimated the risk was 60.2% (Table 3). Knowledge of the guidelines was not associated with better risk assessment (Appendix A).

## 4. Discussion

### 4.1. Main Findings

Our study shows that, despite a minority of midwives reporting an awareness of the 2017 CNGOF guidelines, the majority of them adhere to these guidelines in their diagnoses and provision of care. However, and fortunately without affecting the medical responses, a significant proportion of midwives are inclined to underestimate the risks to newborns in cases of initial infection at term. There is a lack of knowledge about the new definitions of primary and nonprimary initial infections, with no impact on practise.

### 4.2. Interpretation

Worryingly, our first finding showed that only a quarter of midwives claimed to be aware of the national guidelines 3 years following their publication. This result is comparable to those found in other sectors where members of organizations fail to integrate new practises into their routines [13,14]. In addition, it is widely accepted that it may take up to 17 years for the guidelines to be fully integrated into current practises [15]. These elements could explain why these new guidelines have not been fully integrated into clinical practise 3 years following their publication. Indeed, we should not underestimate the impact of opinion leaders. They can directly impact education and the promotion and adoption of new guidelines in their institutions. We could use local opinion leaders to accelerate the successful adoption of new practises [16,17].

Reassuringly, the majority of midwives adopted practises consistent with the guidelines even though they said they had not heard of them. We observed the same results as in the study conducted by Hegarty et al., which was conducted before the publication of the guidelines [10]. In the present study, more than 70% of the midwives recommended performing a caesarean delivery to reduce transmission to the newborn in case of a lesion during labour with intact membranes. If the decision on the mode of birth at term was mostly inappropriate, this has no impact on the health of the newborns because, in the French context, midwives faced with this situation must call a physician for medical advice [18]. However, a simple diffusion of the new guidelines does not change the practises. The same trend has been observed in other areas, such as in the management of postpartum haemorrhaging [19,20]. It therefore seems necessary to consider new strategies, such as the clinical spotlight [21].

We noted an underestimation of neonatal risk in the case of nonprimary initial infection. Once again, this underestimation does not appear to be associated with a lesser familiarity with the guidelines. This is worrying because a better estimation of neonatal risk could lead to better information for parents, and appropriate information is likely to improve compliance, therapeutic compliance, and positive health outcomes [22].

### 4.3. Strengths and Limitations

The main strength of this study is that it was conducted on a national scale. However, we found an underrepresentation of midwives practicing in nonhospital environments. This could be explained by the fact that these midwives are not necessarily concerned by the guidelines. The study’s main limitation is probably the selection bias inherent in internet surveys and the similarly inherent social desirability bias. However, our sample included midwives of different genders and ages and from different regions. Considering that about one-third of the participants did not complete the questionnaire, we can assume that a shorter questionnaire would have improved the rate of response.

### 4.4. Implications

The main implications of the results for clinical practise is to confirm that the simple dissemination of guidelines is not sufficient to change practises. To improve practise, other interventions, such as multifaceted interventions and simulations, should probably be considered [23,24,25]. Regarding the provision of information to parents, we would need to consider the creation of appropriate and easily accessible materials. We also need to rethink the place of primary care, where midwives have an essential role to play [26,27]. The search for a vaccine must also continue [28,29].

## 5. Conclusions

The majority of reported practises were compliant despite a low level of knowledge of the guidelines. The dissemination of guidelines may be important to improve the evaluation of neonatal risk, information, the informed consent of women, and good adherence to drug therapy.

## Figures and Tables

**Figure 1 healthcare-11-00364-f001:**
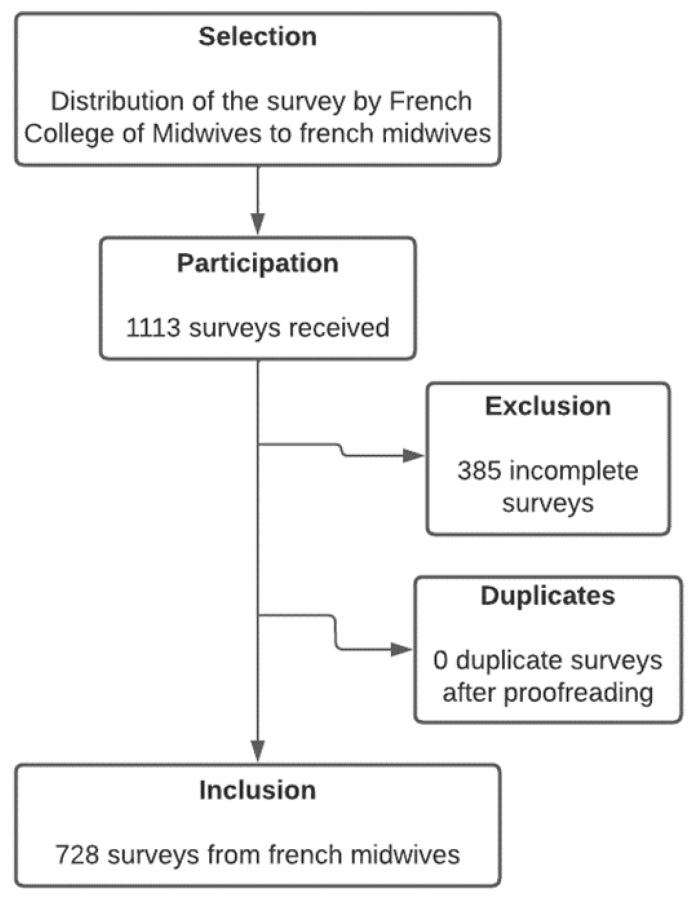
Flowchart.

**Table 1 healthcare-11-00364-t001:** Characteristics of the sample (*n* = 728).

Characteristics	*n*	(%)
Age, median	34	
Gender		
Women	705	96.8
Men	23	3.2
Practise mode		
Nonhospital midwife	43	5.9
Midwife in hospital <1000 births	68	9.3
Midwife in hospital 1000–3000 births	371	51.0
Midwife in hospital >3000 births	246	33.8
Years of practise, median	10 years	
Region		
Ile de France	166	22.8
Auvergne Rhône Alpes	195	26.8
Hauts de France	25	3.4
PACA	27	3.7
Normandie	29	4.0
Grand Est	54	7.4
Occitanie	36	4.9
Nouvelle Aquitaine	72	9.9
Centre Val de Loire	19	2.6
Bourgogne-Franche-Comté	20	2.7
Bretagne	36	4.9
Corse	7	1.0
Pays de la Loire	22	3.0
DOM TOM	17	2.3
Unknown	3	0.4
Knowledge of guidelines	190	26.1

**Table 2 healthcare-11-00364-t002:** Results of clinical scenarios of primary initial infections.

Context of the Clinical Vignette	Number (Percentage) of Answers According to the Term of the Pregnancy	
Primary Initial Infection	25 Weeks of Gestation	38 Weeks of Gestation
Diagnosis	*n* (%)	*n* (%)
Primary initial	693 (95.2) *	708 (97.2) *
Nonprimary initial	20 (2.7)	8 (1.2)
Recurrence	2 (0.3)	3 (0.4)
Don’t know	13 (1.8)	9 (1.2)
Treatment		
Prophylactic	425 (58.4) *	278 (38.2)
Curative	692 (95.1) *	555 (76.2) *
No treatment	3 (0.4)	5 (0.6)
Mode of birth		
Caesarean section at term	19 (2.6)	364 (50.0) *
Vaginal birth at term	622 (85.4) *	9 (1.2)
Don’t know	87 (11.9)	355 (48.8)
Neonatal risk		
High risk	19 (2.6)	374 (51.4) *
Low risk	319 (43.8) *	207 (28.4)
No risk	295 (40.5) *	39 (5.4)
Don’t know	95 (13.1)	108 (14.8)
Breastfeeding		
Yes	727 (99.9) *	704 (96.7) *
No	1 (0.1)	24 (3.3)
Overall conformity	(41.5)	(18.4)

*: correct answer.

**Table 3 healthcare-11-00364-t003:** Results of clinical scenarios of nonprimary initial infections.

Context of the Clinical Vignette	Number (Percentage) of Answers According to the Term of the Pregnancy	
Nonprimary Initial Infection	25 Weeks of Gestation	38 Weeks of Gestation
Diagnosis		
Primary initial	84 (11.5)	110 (15.1)
Nonprimary initial	415 (57.0) *	345 (47.4) *
Recurrence	192 (26.4)	246 (33.8)
Don’t know	37 (5.1)	27 (3.7)
Treatment		
Prophylactic	510 (70.1) *	363 (49.9)
Curative	661 (90.8) *	500 (68.8) *
No treatment	2 (0.2)	2 (0.3)
Mode of birth		
Caesarean section at term	13 (1.8)	257 (35.3) *
Vaginal birth at term	580 (79.7) *	5 (0.7)
Don’t know	135 (18.5)	466 (64.0)
Neonatal risk		
High risk	14 (1.9)	290 (39.8) *
Low risk	354 (48.6) *	268 (36.8)
No risk	233 (32.0) *	45 (6.2)
Don’t know	127 (17.5)	125 (17.2)
Breastfeeding		
Yes	726 (99.7) *	714 (98.1) *
No	2 (0.3)	14 (1.9)
Overall conformity	(29.1)	(5.4)

*: correct answer.

**Table 4 healthcare-11-00364-t004:** Results of clinical scenarios of recurrences.

Context of the Clinical Vignette	Number (Percentage) of Answers According to the Term of the Pregnancy	
Recurrence	25 Weeks of Gestation	38 Weeks of Gestation
Diagnosis		
Primary initial	18 (2.5)	4 (0.5)
Nonprimary initial	47 (6.5)	52 (7.1)
Recurrence	647 (88.8) *	659 (90.5) *
Don’t know	16 (2.2)	13 (1.8)
Treatment		
Prophylactic	531 (72.9) *	442 (60.7)
Curative	547 (75.1) *	409 (56.1) *
No treatment	33 (4.5)	16 (2.2)
Mode of birth		
Caesarean section at term	12 (1.7)	161 (22.1) *
Vaginal birth at term	581 (79.8) *	15 (2.1) *
Don’t know	135 (18.5)	552 (75.8)
Neonatal risk		
High risk	10 (1.4)	129 (17.7)
Low risk	253 (34.8) *	368 (50.6) *
No risk	364 (50.0) *	97 (13.3)
Don’t know	101 (13.8)	134 (18.4)
Breastfeeding		
Yes	723 (99.3) *	71 (98.2) *
No	5 (0.7)	13 (1.8)
Overall conformity	(49.7)	(8.7)

*: correct answer

## Data Availability

The data presented in this study are available on request from the corresponding author.

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
