# Peer review of "Evaluation of Midwives’ Practises on Herpetic Infections during Pregnancy: A French Vignette-Based Study"

_healthcare, 2023, doi:10.3390/healthcare11030364_

Round 1

Reviewer 1 Report

The manuscript is very interesting. The establishment of guidelines is very useful to systematize the clinical practice of nurses and midwives. Although the reality is that, in many cases, these guidelines are unknown or little used. Deepening the knowledge and use of this specific guideline for the diagnosis and treatment of herpes can be very useful to verify the good practices of midwives in these situations.

The introduction frames the study topic very well and the methodology used is adequate.

Overall, it seems to me to be a good manuscript, although I suggest that the authors reorganize the results by inserting the figures and tables in the text after the description of each one of them.

Discussion. The paragraph from lines 161 to 168 would be placed at the end of the discussion in a specific section dedicated to the Limitations of the study.

I believe that the discussion with the findings found should be developed a little more.

Author Response

Thank you very much for the complimentary and constructive comments. As recommended by the reviewer, we have reorganised the results to interleave the figures and tables with the text.

We have also followed the reviewer's suggestions by moving the paragraph from lines 161 to 168 to the end of the discussion and adding the subtitle “Strengths and Limitations”.

Finally, we have expanded the discussion and used more recent references.

Reviewer 2 Report

Midwives' guidance to pregnant women regarding herpesvirus infection is not implemented in all countries, and this role is more likely to be played by obstetricians. In addition, it may be important for the government to take action (government-led intervention by midwives) regarding measures to prevent herpesvirus infection.

The efforts in France are very important, and it is hoped that midwives will improve their skills in the future. In addition, the importance of this study is understandable. Please confirm the following comments.

I think it is necessary to itemize the survey content in the method.

I think it is unnecessary to write one brief word about measures to protect personal information for web-based surveys, and to explain cookies and IP addresses.

Each table should be placed immediately after the paragraph.

What is the significance of the italicized numbers in the tables?

The % in Table1 has no decimal point. They are in the text (e.g., 26.1%). Uniformity of notation is needed.

If the survey does not state in the method that there are two scenarios with different number of weeks of gestation, the results cannot be read.

Is it possible to prepare a document with both the questions and the results, since the correspondence between the results and the survey content is difficult to read as a paper?

I have no objection to the discussion of the results.

Author Response

We sincerely thank the reviewer for his positive and constructive comments which helped us to improve the manuscript of this article.

We have further detailed the content of the survey in the method section, including a description of the clinical vignettes. The complete survey form is now available in the appendix. Although we agree with the reviewer's comment, we have maintained the cookie and IP address information as part of the CHERRIE checklist.

As recommended by the reviewer, we have reorganised the results to interleave the figures and tables with the text.

The italicised items in the tables correspond to the raw responses (i.e. correct answers) to the clinical vignette under consideration. We have moved this note to the bottom of the table and replaced the italics with asterisks for better understanding.

We have standardised the tables so that the percentages are always accurate to one decimal place.

We have specified the number and diagnosis of scenarios (clinical vignettes) in the method section. Finally, we have included the results in Appendix S1 so as to have the correspondence between the results and the survey content as recommended by the reviewer.

Reviewer 3 Report

Dear Authors

The paper presented to me for review raises important issues from the point of view of the quality of care for a pregnant woman.

Detailed notes below:

• a clear and specific goal of the work

• material and method sufficiently described

• results presented extensively with a detailed description

• conclusions that are fit for purpose

The selection of bibliography requires significant improvement (nearly half of the sources are older than 10 years) - I recommend using the current literature. For this reason, the discussion and introduction should also be corrected

Author Response

We sincerely thank the reviewer for his positive and constructive comments which helped us to improve the manuscript of this article. As suggested by the reviewer, we have revised the introduction and discussion to include more recent references.

Reviewer 4 Report

Dear Authors 

This paper is very interesting! However, you may pay attention to some points. 

There is no at the end of the introduction the purpose of your study. 

The methodology section should clearly show the midwife response rate and the final sample. 

The second paragraph of the discussion would be good to move to the end of the section. 

Author Response

We thank the reviewer for his/her appreciation and comments.

The reviewer's comments helped us to improve the paper, in particular by specifying the main objective of our study at the end of the introduction.

We have added a sentence in the method to specify the number of hospital midwives in France at the time of our study

We have also followed the reviewer's suggestions by moving the second paragraph of the discussion to the end of the discussion and adding the subtitle “Strengths and Limitations”.

Round 2

Reviewer 3 Report

Thank you very much for considering my suggestions